# Lifetime homelessness among young transgender women in Lima, Peru is associated with HIV vulnerability: Results from a cross-sectional survey

**Dorothy Apedaile**[1]*, **Alfonso Silva-Santisteban**[2], **Sari L. Reisner**[3,4], **Leyla Huerta**[5], **Segundo R. León**[6], **Amaya Perez-Brumer**[1,2]

**1** Dalla School of Public Health, University of Toronto, Toronto, Canada, **2** Center for Interdisciplinary Research in Sexuality, AIDS and Society, Universidad Peruana Cayetano Heredia, Lima, Peru, **3** Department of Epidemiology, University of Michigan School of Public Health, Ann Arbor, Michigan, United States of America, **4** Department of Epidemiology, Harvard T.H. Chan School of Public Health, Boston, Massachusetts, United States of America, **5** Féminas, Lima, Peru, **6** Escuela Profesional de Tecnología Médica, Universidad Privada San Juan Bautista, Lima, Peru

* Dorothy.apedaile@mail.utoronto.ca

## Abstract

Transgender youth face disproportionately high rates of homelessness, which can increase vulnerability to HIV. In Peru, the incidence of HIV among transgender women has increased 19% since 2010 and young transgender women are a priority population for HIV prevention. We sought to estimate the proportion of young transgender women experiencing homelessness and associations between homelessness and HIV vulnerabilities. We recruited transgender women ages 16–24 years (N = 209) to participate in a biobehavioural survey and HIV and STI testing (chlamydia, syphilis, gonorrhea). Poisson regression models with robust standard errors were fit to estimate the association between past homelessness and past 6-month condomless sex, adjusted for potential confounding by age, education, sex work, non-injection drug use, post-traumatic stress disorder (PTSD), and violence. Among participants (median age 23 years), 68 (32.5%) had ever been homeless and 19 (9.1%) reported homelessness in the past 3 months. Overall, 51.5% of those who had been homeless reported past 6-month condomless sex compared to 29.1% of those who had never been homeless (p < 0.001). HIV prevalence was 44.6% among those with a history of homelessness and 39.6% among those who had never been homeless (p = 0.65); lifetime homelessness was significantly associated with increased sex work (p < 0.001), violence (p < 0.01), and PTSD (p < 0.001). In the model adjusting for age, education, and behavioural risk (sex work, non-injection drug use), participants who had been homeless had 1.43 times higher prevalence of past 6-month condomless sex (95% CI = 1.05-1.96); results were attenuated when adjusting for violence and PTSD. The high prevalence of homelessness among young transgender women sampled underscores the compounding HIV vulnerabilities faced by this population. Efforts to prevent homelessness and improve access to housing are urgently needed alongside

**Data availability statement:** The data underlying the findings in the manuscript contain potentially identifying and sensitive patient information and are not available due to ethical and legal restrictions from the research ethics board at Universidad Peruana Cayetano Heredia (UCPH). Data will be made available upon reasonable request to the corresponding author or UCPH Research Ethics Board (orvei.ciei@oficinas-upch.pe). To maintain long-term data availability and security, all data is stored by the study principle investigators on a secure, cloud-based platform that adheres to institutional and ethical data protection standards in non-proprietary formats (e.g., CSV) in addition to a secure back up by the corresponding author.

**Funding:** This work was funded by the National Institute of Mental Health of the National Institutes of Health (R21MH118110 to SLR and AS-S) and Canadian Institutes of Health Research (CRC-2021-00132 to AP-B). The funders had no role in study design, data collection and analysis, decision to publish, or preparation of the manuscript. The content is solely the responsibility of the authors and does not necessarily represent the official views of the funding agencies.

**Competing interests:** The authors have declared that no competing interests exist.

healthcare services, including HIV prevention and treatment, to address the disproportionate HIV epidemic among young Peruvian transgender women.

## Introduction

Transgender young people ages 16–24 experience disproportionately high rates of homelessness compared to their non-transgender peers, based on studies from North America [1,2]. Both familial rejection and violence as well as systemic exclusion can contribute to high rates of homelessness among transgender youth [3,4]. While most definitions of youth homelessness focus on youth who are living independently from their parents or guardians, it is also possible for transgender youth to experience homelessness alongside members of their families [5,6]. Once homeless, transgender youth face additional barriers to accessing support services and finding new housing and this can have lasting health consequences [2,7,8]. However, limited research has investigated homelessness among transgender people outside of North America. A 2019 systematic review of the experiences and healthcare needs of homeless LGBTQ youth found only studies from Canada and the United States and few focussed on transgender youth only [7]. A review of homelessness among transgender people similarly found no studies from outside North America [9].

The lack of research about transgender youth homelessness outside North America is particularly concerning in the context of consistently high HIV incidence among youth from key populations, including transgender women [10]. Homelessness can increase vulnerability to HIV and other sexually transmitted infections (STIs) through decreased access to healthcare services as well as increased biopsychosocial vulnerabilities through sex work and violence [11–13]. People experiencing homelessness face discrimination from healthcare providers as well as logistical barriers to accessing healthcare services [14,15]. A lack of health insurance and gender-congruent identification documents can further limit access to health care, particularly for transgender people [16]. There is increasing attention to the importance of addressing homelessness to prevent HIV and improve the health and wellbeing of people living with HIV, particularly for populations that already face a high burden of HIV, such as transgender women [17,18]. Stable housing is strongly associated with better adherence to HIV treatment and improved health outcomes for people living with HIV [19]. In contrast, homelessness and housing instability are associated with decreased uptake and adherence to HIV prevention strategies, such as pre-exposure prophylaxis (PrEP) [20,21].

Homelessness can also decrease the ability to negotiate condom use with paying and non-paying sexual partners in the context of economic precarity (e.g., being offered more money to engage in condomless sex), particularly for individuals who are relying on friends, intimate partners, or sex work clients for temporary housing [22,23]. Condomless sex is a key pathway for HIV acquisition and transmission among transgender women [24]. Importantly, poor mental health, substance use, and interpersonal violence can all contribute to decreased condom use as well as increased homelessness [25,26].

In Latin America and the Caribbean, there were an estimated 11,000 new cases of HIV among adolescents ages 10-19 years in 2023 and HIV incidence in Latin America has increased 19% among transgender women since 2010 [27,28]. This rising incidence occurs in contexts where young transgender women face pervasive violence and marginalization, including harassment and discrimination from state actors (e.g., police), difficulty finding employment, high rates of poverty, and high rates of violence [16,29,30]. Further, studies conducted in Latin America have found that transgender women aged 18-24 are at a higher risk of HIV incidence and less likely to engage in HIV prevention strategies such as condom use and

PrEP [31,32]. In Peru, the estimated prevalence of HIV among transgender women is 29.8%-48.8% compared to 0.4% prevalence in the general population of Peru [33,34]. Consequently, young transgender women ages 16-24 years are a critical group for primary HIV prevention efforts in Peru, and further information on their social vulnerabilities is crucial to informing efforts to reduce HIV incidence, improve health outcomes for those living with HIV, and improve health and wellbeing more broadly. The purpose of this exploratory analysis was to examine the prevalence of lifetime homelessness among young transgender women in Lima, Peru and associations between lifetime homelessness and HIV vulnerability.

## Methods

### Study design and participants

From February to July 2022, we conducted a cross-sectional survey and testing for HIV and bacterial STIs (gonorrhea, chlamydia, and syphilis) among transgender women ages 16-24 years in Lima, Peru. Eligibility criteria included being ages 16-24 years, being a transgender woman (assigned a male sex at birth and identifying on the transfeminine continuum [35]), and living in Lima, Peru. The study design was informed by qualitative research and consultation with Féminas, a community-based organization formed and led by transgender women [16]. Participants were recruited by peers and recruiters from a community organization and surveys were administered by peer survey interviewers. Study participants were also invited to encourage potential participants from their social networks to contact the study team. Detailed study procedures have been published elsewhere [36]. Of the 211 young transgender women who participated in the study, this analysis is restricted to the 209 participants with complete responses to the survey questions regarding homelessness.

### Ethics statement

All study activities were approved by the Universidad Peruana Cayetano Heredia Institutional Committee on Research Ethics. Written informed consent (participants 18 or older) or assent (participants ages 16 to 17) was obtained from all participants prior to study participation. For participants ages 16 to 17, parental consent was waived when it would potentially result in harm to the participant.

### Measures

Measures were drawn from prior research fielded with transgender populations generally or transgender women specifically [37–39].

**Primary exposure: Lifetime homelessness.** The primary exposure, lifetime homelessness, was measured as a binary variable representing whether the participant had been homeless at any point. Homelessness was defined as sleeping in a shelter, on the street, in a car, in some other place not designed for sleeping, or in the home of a friend or relative for a few nights or weeks. Participants were also asked if they had been homeless in the past 30 days and 3 months as well as their current living situation.

**Outcome: Past 6-month condomless sex.** The primary outcome was any condomless sex in the past 6 months with a partner living with HIV or a partner with an unknown HIV status. This binary variable was derived from survey questions where participants were asked how many times they had sex (insertive or receptive vaginal or anal sex) in the past 6 months, how many of those times were without a condom, and the HIV status of their partners.

**Sociodemographics.** Age, place of birth, educational attainment, ever dropped out of school, and monthly household income were assessed. Employment status was assessed as formal employment (full time or part-time job with a salary), informal work, or

unemployment. Food insecurity was defined as participants reporting they sometimes, most of the time, or almost always ran out of food or money to buy food by the end of the month.

**HIV and STI vulnerability, sex work, and HIV prevention.** Sex work was defined as engaging in sex work ever and in the past 30 days. History of HIV testing was queried. Participants were asked if they had ever used PrEP and, after explaining PrEP, if they were willing to use daily oral PrEP.

**Psychosocial vulnerabilities.** Gender affirmation was assessed based on lifetime hormone use. Lifetime experiences of violence were assessed by asking about physical, psychological, or sexual violence; lifetime intimate partner violence was measured using the Trans-specific Intimate Partner Violence Scale [40] (e.g., "Did your partner tell you or threaten to tell someone else that you are transgender against your will, in order to humiliate you or to make you feel unsafe?"); and past-year sexual objectification was measured using an adapted version of the Spanish-language Interpersonal Sexual Objectification Scale [41]. Participants were defined as lacking family acceptance if they disagreed or strongly disagreed that their families accepted and supported their gender identity. The questionnaire also assessed psychological distress using the validated six-item Kessler-6 psychological distress tool (score >= 13) [42], post-traumatic stress disorder using the five-item Primary Care PTSD Screen for DSM-V post-traumatic stress disorder symptoms used in prior transgender health research (score >= 4) [43], suicidal ideation and attempts, non-injection drug use (e.g., marijuana, benzodiazepines, hallucinogens) and alcohol misuse (using the AUDIT-C score >= 3 from the Spanish-language version of the AUDIT-C) [44].

**Healthcare.** Health insurance, usual source of medical care, and last visit to a medical provider were measured. Health insurance options included public insurance available to individuals living in poverty; private insurance paid for by the participant, their employer, or their family member; or no health insurance. Anticipated discrimination from a healthcare provider was assessed using the healthcare item from the Intersectional Discrimination Index–Anticipated discrimination [45] and barriers to healthcare access were measured by asked participants to indicate which challenges they faced when seeking healthcare.

## Laboratory procedures

After completing the survey, participants were offered testing for HIV, syphilis, chlamydia, and gonorrhea. HIV testing was performed by using two rapid HIV tests ([Alere Determine™ HIV-1/2 Ag/Ab Combo—Alere, Waltham, MA, USA] and SURE CHECK® HIV 1/2 Assay [Chembio Diagnostic Systems Inc, NY, USA]) in parallel. Pre-test counselling was provided by a certified HIV test counsellor following Peruvian guidelines. Confirmatory testing was performed via a combination of regular enzyme immunoassay and Western blot (Genscreen ULTRA HIV Ag-Ab Assay and NEW LABBLOT HIV-1, BioRad, France). Those diagnosed as HIV positive were referred to the National Antiretroviral Treatment (Programa TARGA). Syphilis testing included qualitative and quantitative Rapid Plasma Reagin (Syphilis RPR Test, Human Diagnostics, Germany) tests followed by confirmation through Treponema pallidum-Particle Agglutination test (Syphilis TPHA liquid, Human Diagnostics, Germany) using a cut-off value of 1:80; a threshold of RPR titers >1:8 was used to identify syphilis. A pharyngeal swab was also collected for chlamydia and gonorrhea testing using the Aptima Combo 2® Assay (Hologic, San>Diego, USA). Participants were classified as having a bacterial STI if they had a positive test result for chlamydia, gonorrhea, or syphilis.

## Statistical analysis

Descriptive statistics were used to characterize participants overall and by lifetime experience of homelessness. Chi-square tests were used to compare young transgender women who had been homeless to those who have never been homeless for categorical variables and Kruskall-Wallis tests were used for continuous variables, excluding participants with missing data. For categorical variables with expected cell values ≤5, Fisher's exact tests were used. Sensitivity analyses for these comparisons were conducted including the missing data level as an explicit category. Cronbach's alpha was estimated for all continuous scales to assess reliability. Poisson regression models with robust standard errors were fit to estimate the association between lifetime homelessness and past 6-month condomless sex in unadjusted and adjusted models with covariates added in blocks [46,47]. In the first model, we adjusted for sociodemographic characteristics (age and education). In the second model, we adjusted for sociodemographic characteristics and key behavioural risks that can increase engagement in condomless sex (sex work and non-injection drug use). In the third model, we adjusted for sociodemographic characteristics, behavioural risks, and psychosocial vulnerabilities that can increase engagement in condomless sex as well as homelessness (PTSD and violence). For the regression models, participants were excluded if they were missing data on the outcome or any covariates, resulting in an analytical sample of 176 participants. A sensitivity analysis was conducted where homelessness was operationalized as a categorical variable (homeless in the past 3 months, lifetime homelessness, never homelessness). Statistical significance was predetermined at the alpha 0.05 level. All analyses were conducted in R.

## Results

### Sample characteristics

A total of 209 young transgender women were included in this analysis. The median age was 23 years (IQR = 21-24) and 44.5% reported being born in Lima (Table 1). Overall, 68 (32.6%) reported experiencing homelessness in their lifetime, with 19 (9.1%) reporting homelessness in the past 3 months. Young transgender women mostly reported their current housing status as living in their own house, apartment, or room (34.9%) or living with family (27.3%). In addition, 11.5% reported living in a *"casa para personas trans"* (collective house for transgender women). However, those who had never been homeless were more likely to report currently living with family (31.2% vs 19.1%) and less likely to report living in their own rented house, apartment, or room (28.4% vs 48.5%). Participants reported high socio-economic vulnerability, with 45.0% reporting less than 500 soles (~130 USD) per month in income, less than half of the monthly minimum wage, and 65.1% reporting food insecurity. While there was no difference in the highest level of education completed for participants with and without a history of homelessness, 55.9% of participants who had been homeless reported dropping out of school, compared to 28.4% of those who had never been homeless.

### Homelessness, HIV and bacterial STI vulnerability, sex work, and HIV prevention

Table 2 presents HIV and STI vulnerability variables comparing young transgender women with and without experiences of homelessness. Overall, 67 (41.4%) participants tested positive for HIV and 74 (46.2%) tested positive for a bacterial STI. There were no significant differences in the prevalence of HIV among those with and without a history of homelessness. Among the 67 trans women testing HIV seropositive, 11 (16.4%) had been homeless in the past 3 months and another 14 (20.9%) had experienced homelessness more than 3 months

**Table 1. Sociodemographic characteristics and experiences of precarity among young transgender women in Lima, Peru.**

| | Overall N = 209 n (%) | Ever homeless N = 68 n (%) | Never homeless N = 141 n (%) | P-value |
|---|---|---|---|---|
| **Age in years (median [IQR])** | 23.0 [21.0, 24.0] | 23.00 [21.00, 24.00] | 23.0 [21.0, 24.0] | 0.449 |
| **Place of birth** | | | | 0.100 |
| Lima | 93 (44.5) | 27 (39.7) | 66 (46.8) | |
| Other regions of Peru | 93 (44.5) | 30 (44.1) | 63 (44.7) | |
| Other countries[1] | 16 (7.7) | 9 (13.2) | 7 (5.0) | |
| Missing | 7 (3.3) | 2 (2.9) | 5 (3.5) | |
| **Highest level of education** | | | | 0.774 |
| Less than secondary school | 57 (27.3) | 18 (26.5) | 39 (27.7) | |
| Secondary school | 76 (36.4) | 23 (33.8) | 53 (37.6) | |
| Post-secondary | 76 (36.4) | 27 (39.7) | 49 (34.8) | |
| Missing | 0 (0.0) | 0 (0.0) | 0 (0.0) | |
| **Ever dropped out of school** | | | | <0.001 |
| Yes | 78 (37.3) | 38 (55.9) | 40 (28.4) | |
| No | 131 (62.7) | 30 (44.1) | 101 (71.6) | |
| Missing | 0 (0.0) | 0 (0.0) | 0 (0.0) | |
| **Employment status** | | | | 0.051 |
| Formally employed | 24 (11.5) | 9 (13.2) | 15 (10.6) | |
| Informal employment | 112 (53.6) | 43 (63.2) | 69 (48.9) | |
| Unemployed | 70 (33.5) | 15 (22.1) | 55 (39.0) | |
| Missing | 3 (1.4) | 1 (1.5) | 2 (1.4) | |
| **Monthly household income** | | | | 0.190 |
| ≤500 soles | 94 (45.0) | 28 (41.2) | 66 (46.8) | |
| 501 – 1500 soles | 50 (23.9) | 22 (32.4) | 28 (19.9) | |
| ≥1501 soles | 31 (14.8) | 9 (13.2) | 22 (15.6) | |
| Missing | 34 (16.3) | 9 (13.2) | 25 (17.7) | |
| **Current housing** | | | | 0.001 |
| House, apartment, or room | 73 (34.9) | 33 (48.5) | 40 (28.4) | |
| Parent/relative's home | 57 (27.3) | 13 (19.1) | 44 (31.2) | |
| Intimate partner's home | 27 (12.9) | 3 (4.4) | 24 (17.0) | |
| House for trans women | 24 (11.5) | 9 (13.2) | 15 (10.6) | |
| Friend's home | 19 (9.1) | 6 (8.8) | 13 (9.2) | |
| Street or abandoned building | 3 (1.4) | 3 (4.4) | 0 (0.0) | |
| Other[2] | 2 (1.0) | 1 (1.5) | 1 (0.7) | |
| Missing | 4 (1.9) | 0 (0.0) | 4 (2.8) | |
| **Ever run out of food by end of month** | | | | 0.231 |
| Yes | 136 (65.1) | 50 (73.5) | 86 (61.0) | |
| No | 63 (20.1) | 17 (25.0) | 46 (32.6) | |
| Missing | 10 (4.8) | 1 (1.5) | 9 (6.4) | |

[1]Other countries: Venezuela (n = 15), Spain (n = 1; experienced homelessness ).

[2]Other current housing: hotel (n = 1), hospital or medical establishment (n = 1).

Tests for significance estimated with missing data excluded using global chi-square test; Kruskall-Wallis test for non-normal data (age), and Fisher's exact test for variables with cell ≤5.

**Table 2. HIV and bacterial STI vulnerability, sex work, and HIV prevention among young transgender women with and without experiences of homelessness.**

| | Ever homeless N = 68 n (%) | Never homeless N = 141 n (%) | P-value |
|---|---|---|---|
| **HIV status[1]** | | | 0.65 |
| Positive | 25 (44.6) | 42 (39.6) | |
| Negative | 31 (55.4) | 64 (60.4) | |
| **Bacterial STI[2]** | | | 1.00 |
| Positive | 23 (44.2) | 51 (47.2) | 0.85 |
| Negative | 29 (55.8) | 77 (53.5) | |
| **Sexual partners in past 6 months (median, IQR)[3]** | 22.00 [5.75-120.50] | 3.00 [0.00-90.00] | 0.001 |
| **Condomless sex with HIV positive or status unknown partners in past 6 months** | | | <0.001 |
| Yes | 35 (51.5) | 41 (29.1) | |
| No | 21 (30.9) | 83 (58.9) | |
| Missing | 12 (17.6) | 17 (12.1) | |
| **Ever engaged in sex work** | | | 0.001 |
| Yes | 58 (85.3) | 87 (61.7) | |
| No | 10 (14.7) | 53 (37.6) | |
| Missing | 0 (0.0) | 1 (0.7) | |
| **Engaged in sex work in the past 30 days** | | | 0.033 |
| Yes | 43 (63.2) | 64 (45.4) | |
| No | 25 (36.8) | 74 (52.5) | |
| Missing | 0 (0.0) | 3 (2.1) | |
| **Ever tested for HIV prior to study** | | | 0.016 |
| Yes | 50 (73.5) | 77 (54.6) | |
| No | 18 (26.5) | 63 (44.7) | |
| Missing | 0 (0.0) | 1 (0.7) | |
| **Ever used PrEP[4]** | | | 0.220 |
| Yes | 6 (14.0) | 6 (6.4) | |
| No | 34 (79.1) | 86 (91.5) | |
| Missing | 2 (2.1) | 2 (2.1) | |
| **Willing to use PrEP[4]** | | | 0.058 |
| Yes | 28 (65.1) | 40 (42.6) | |
| No | 14 (32.9) | 45 (47.9) | |
| Missing | 1 (2.3) | 9 (9.6) | |

[1]Among n = 162 participants who completed HIV testing.

[2]Among n = 160 participants with complete tests for syphilis (TPHA positive, RPR > 1:8), gonorrhea (pharyngeal), or chlamydia (pharyngeal).

[3]Excluding 20 participants missing data on number of sexual partners (n = 16 among the never homeless group, n = 4 among the ever homeless group).

[4]Among n = 137 HIV negative or status unknown participants.

Tests for significance estimated with missing data excluded using global chi-square test; Fisher's exact test for variables with cell ≤5.

ago (S1 Table). Young transgender women who had experienced homelessness reported a higher number of sexual partners in the past 6 months (median of 22 partners versus 3; p = 0.001) and were more likely to report lifetime and recent sex work. However, they were also more likely to have been tested for HIV and 65.1% were willing to use daily oral PrEP compared to 42.6% of those who had never been homeless (p = 0.058). Participants with a history of homelessness were more likely to report condomless sex with a partner living with HIV or status unknown partner in the past 6 months (51.5%) compared to those who had never been homeless (29.1%). Among participants with complete data on condomless sex, 43.8% of participants who had been homelessness in the past 3 months reported condomless sex in the past 6 months (Fig 1).

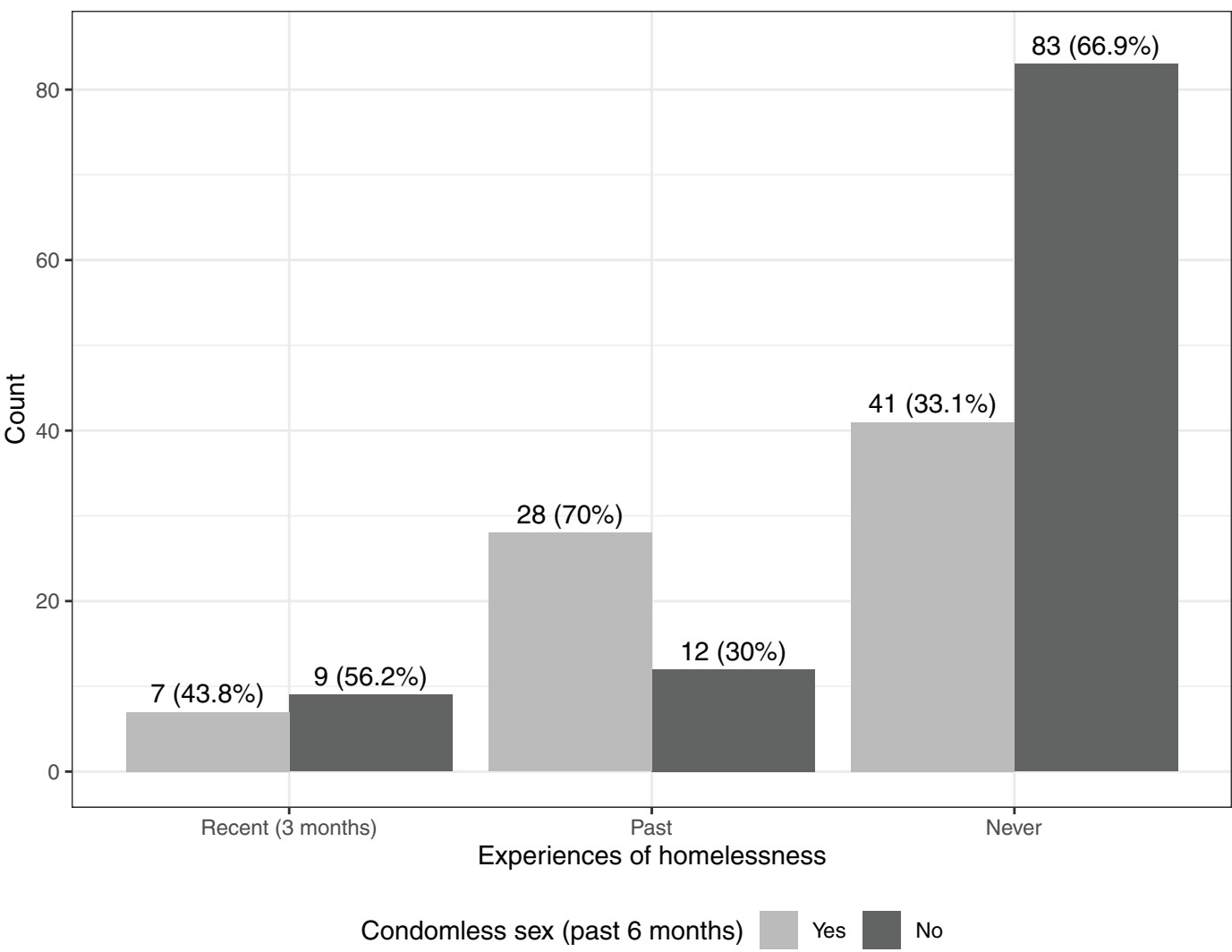

**Fig 1. Number of participants reporting condomless sex in the past 6 months, stratified by experiences of homelessness (recent homelessness (past 3 months), lifetime homelessness (more than 3 months ago), never experiencing homelessness). Note**: Only participants with complete data on condomless sex (n = 180) were included in this figure.

### Homelessness and psychosocial vulnerabilities

Young transgender women who had experienced homelessness reported higher rates of hormone use (Table 3). They were also significantly more likely to report lifetime physical, psychological, and sexual violence, as well as lifetime trans-specific intimate partner violence and high levels of sexual objectification in the past year. While transgender women who had previously been homeless reported similar rates of family acceptance (58.8%) to those who had never been homeless (53.9%), a history of homelessness was associated with experiencing a higher number of adverse childhood experiences (ACEs; median of 3 ACEs vs 4, p=0.03). Participants who had been homeless were more likely to endorse PTSD (35.3%) and current psychological distress (33.8%) and were also more likely to report attempting suicide and suicidal thoughts than young transgender women who had never experienced homelessness. The trans-specific intimate partner violence scale (α = 0.94), interpersonal sexual objectification scale (α = 0.95), primary care PTSD screen for DSM-V (α = 0.93) and Kessler 6-item psychological distress scale (α = 0.88) displayed good to excellent reliability while the AUDIT-C scale displayed lower reliability (α = 0.69).

### Homelessness and healthcare access

Table 4 presents healthcare access among transgender women with and without a history of homelessness. Those who had been homeless were more likely to report having no health insurance (27.9% vs 18.4%) or having publicly-funded insurance (51.5% vs 43.3%). Approximately one-third of participants from both groups reported seeing a medical provider in the past year. A higher proportion of those who had been homeless reported anticipating discrimination from a healthcare provider than those without a history of homelessness (45.6% versus 24.8%; p = 0.003). Finally, participants who had been homeless were more likely to endorse logistical barriers to healthcare access, including transportation (60.3% vs 27.0%, p < 0.001), safety going to or from healthcare (47.1% vs 21.3%; p<0.001), and inconvenient hours (44.1% vs 26.2%; p = 0.013).

### Regression models of homelessness and past 6-month condomless sex

The results of multivariable regression models estimating the association between past homelessness and recent HIV risk (condomless sex in the past 6 months with a partner living with HIV or a partner with unknown HIV status) are presented in Table 5. In the model adjusting for sociodemographic characteristics (age, education) and behavioural risk (sex work and non-injection drug use), participants who had previously been homeless had 1.43 times higher prevalence of recent condomless sex than those who had not been homeless (95% CI=1.05-1.96). When variables representing experiences of violence and poor mental health were added to the model, the association between homelessness and condomless sex was attenuated and no longer statistically significant (aPR = 1.19, 95% CI = 0.87-1.63). When conducting the sensitivity analysis with homelessness operationalized as a 3-level variable (recent homelessness, lifetime homelessness, never homeless), the estimates for lifetime homelessness compared to never experiencing homelessness were similar (S2 Table). Smaller effect sizes with wider confidence intervals were observed for the association between recent homelessness and recent condomless sex.

## Discussion

In this cross-sectional study of young transgender women ages 16–24 years living in Lima, Peru, approximately one-third had experienced homelessness at least once in their lifetime, with more than 1 in 10 experiencing homelessness in the last 3 months. Our findings are

Table 3. Psychosocial vulnerabilities among transgender women with and without experiences of homelessness.

| | Ever homeless N = 68 n (%) | Never homeless N = 141 n (%) | P-value |
|---|---|---|---|
| **Gender affirmation** | | | |
| **Hormone use** | | | <0.001 |
| Yes | 48 (70.6) | 57 (40.4) | |
| No | 20 (29.4) | 80 (56.7) | |
| Missing | 0 (0.0) | 4 (2.8) | |
| **Violence** | | | |
| **Physical violence** | | | 0.003 |
| Yes | 41 (60.3) | 52 (36.9) | |
| No | 26 (38.2) | 85 (60.3) | |
| Missing | 1 (1.5) | 4 (2.8) | <0.001 |
| **Psychological violence** | | | |
| Yes | 53 (77.9) | 65 (46.1) | |
| No | 15 (22.1) | 74 (52.5) | |
| Missing | 0 (0.0) | 2 (1.4) | |
| **Sexual violence** | | | <0.001 |
| Yes | 31 (45.6) | 23 (16.3) | |
| No | 37 (54.4) | 118 (83.7) | |
| Missing | 0 (0.0) | 0 (0.0) | |
| **Transgender-specific intimate partner violence** | | | 0.004 |
| Yes | 26 (38.2) | 26 (18.4) | |
| No | 42 (61.8) | 112 (79.4) | |
| Missing | 0 (0.0) | 3 (2.1) | |
| **Sexual objectification (past year)[1]** | | | 0.001 |
| Low (score 0–9 on ISOS) | 23 (33.8) | 81 (57.4) | |
| High (score 10–16 on ISOS) | 45 (66.2) | 57 (40.4) | |
| Missing | 0 (0.0) | 3 (2.1) | |
| **Family and childhood experience** | | | |
| **Family acceptance** | | | 0.760 |
| Yes | 40 (58.8) | 76 (53.9) | |
| No | 28 (41.2) | 61 (43.3) | |
| Missing | 0 (0.0) | 4 (2.8) | |
| **Adverse childhood experiences** | | | |
| Median [IQR] | 4.00 [2.75-6.00] | 3.00 [0.75-5.00] | 0.031 |
| **Mental health and substance use** | | | |
| **Ever attempted suicide** | | | 0.134 |
| Yes | 21 (30.9) | 29 (20.6) | |
| No | 46 (67.6) | 111 (78.7) | |
| Missing | 1 (1.5) | 1 (0.7) | |
| **Ever had suicidal ideation** | | | 0.056 |
| Yes | 29 (42.6) | 41 (29.1) | |
| No | 37 (54.4) | 99 (70.2) | |
| Missing | 2 (2.9) | 1 (0.7) | |
| **Post-traumatic stress disorder (PTSD)** | | | <0.001 |
| Yes (≥4 on PC-PTSD-5) | 24 (35.3) | 18 (12.8) | |
| No (<4 on PC-PTSD-5) | 44 (64.7) | 123 (87.2) | |

*(Continued)*

**Table 3.** (Continued)

| | Ever homeless N = 68 n (%) | Never homeless N = 141 n (%) | P-value |
|---|---|---|---|
| Missing | 0 (0.0) | 0 (0.0) | |
| **Psychological distress** | | | 0.001 |
| Yes (≥13 on K6) | 23 (33.8) | 19 (13.5) | |
| No (<13 on K6) | 44 (64.7) | 120 (85.1) | |
| Missing | 1 (1.5) | 2 (1.4) | |
| **Alcohol misuse** | | | 0.388 |
| Yes (≥3 on AUDIT-C) | 45 (66.2) | 90 (63.8) | |
| No (<3 on AUDIC-C) | 10 (14.7) | 12 (8.5) | |
| Missing | 13 (19.1) | 39 (27.7) | |
| **Non-injection drug use[2]** | | | <0.001 |
| Yes | 39 (57.4) | 41 (29.1) | |
| No | 27 (39.7) | 97 (68.8) | |
| Missing | 2 (2.9) | 1 (0.7) | |

[1]Measured using the Interpersonal Sexual Objectification Scale, possible scores range from 0–16 with higher scores representing more frequent sexual objectification; dichotomized into "high" and "low" categories based on the median value.

[2]Including marijuana use.

Abbr: ISOS: Interpersonal Sexual Objectification Scale; PC-PTSD-5: Primary Care PTSD Screen for DSM-V; K6: Kessler 6-item psychological distress tool; AUDIT-C: Alcohol Use Disorders Identification Test.

Tests for significance estimated with missing data excluded using global chi-square test; Kruskall-Wallis test for non-normal continuous data; Fisher's exact test for variables with cell ≤5.

consistent with the most recent US Trans Survey, which found one-third of transgender adults had experienced homelessness in their lifetime, and results from the 2021 National Survey on LGBTQ Mental Health, which found that 38% of young transgender women ages 13–24 years had experienced homelessness [39,48]. We also found that lifetime homelessness was significantly associated with increased HIV vulnerabilities through engagement in condomless sex with a partner living with HIV or status unknown partner in the past 6 months. This finding corroborates prior research among transgender women and extends it to young transgender women in the Peruvian setting [49]. Although these results were attenuated after adjusting for PTSD and violence, findings point to an important need to tailor HIV prevention strategies for young transgender women who experience homelessness. Young transgender women who had experienced homelessness also reported barriers to healthcare access above and beyond barriers already experienced by young transgender women, including lack of health insurance, anticipated discrimination in healthcare, and logistical barriers to care access. These results suggest that young transgender women in Peru need support to secure and maintain adequate housing, and that addressing housing and other material needs is crucial to HIV prevention efforts for young transgender women.

For transgender women in our sample, homelessness was situated alongside other vulnerabilities.

The young transgender women in this study also reported high rates of material deprivation more broadly, including very low incomes, lack of employment, and high levels of food insecurity. While there is limited research on homelessness in Peru, studies in Brazil have found higher rates of STIs among people experiencing homelessness, high rates of adverse

**Table 4. Healthcare access among young transgender women in Lima, Peru with and without experiences of homelessness.**

|  | Ever homeless N = 68 n (%) | Never homeless N = 141 n (%) | P-value |
|---|---|---|---|
| **Health insurance** |  |  | 0.017 |
| None | 19 (27.9) | 26 (18.4) |  |
| Private | 12 (17.6) | 52 (36.9) |  |
| Public | 35 (51.5) | 61 (43.3) |  |
| Missing | 2 (2.9) | 2 (1.4) |  |
| **Usual source of medical care** |  |  | 0.308 |
| Emergency room | 4 (5.9) | 21 (14.9) |  |
| Medical health post | 41 (60.3) | 72 (51.1) |  |
| Private clinic | 11 (16.2) | 18 (12.8) |  |
| Other | 2 (2.9) | 4 (2.8) |  |
| No source of medical care | 9 (13.2) | 13 (9.2) |  |
| Missing | 1 (1.5) | 13 (9.2) |  |
| **Last visit to a medical provider** |  |  | 0.544 |
| <1 year ago | 43 (63.2) | 93 (66.0) |  |
| ≥1 years ago | 22 (32.4) | 37 (26.2) |  |
| Missing | 3 (4.4) | 11 (7.8) |  |
| **Anticipate discrimination by a healthcare provider[1]** |  |  | 0.003 |
| Yes | 31 (45.6) | 35 (24.8) |  |
| No | 35 (51.5) | 105 (74.5) |  |
| Missing | 2 (2.9) | 1 (0.7) |  |
| **Barriers to healthcare** |  |  |  |
| Availability of time | 37 (54.4) | 58 (41.1) | 0.086 |
| Transportation | 41 (60.3) | 38 (27.0) | <0.001 |
| Safety going to or from health care | 32 (47.1) | 30 (21.3) | <0.001 |
| Childcare | 12 (17.6) | 19 (13.5) | 0.542 |
| Cost | 36 (52.9) | 60 (42.6) | 0.187 |
| No health coverage | 26 (38.2) | 29 (20.6) | 0.010 |
| Inconvenient hours | 30 (44.1) | 37 (26.2) | 0.013 |
| Mistreatment by staff or other patients for being transgender | 23 (33.8) | 34 (24.1) | 0.178 |
| Bad experiences in the past | 23 (33.8) | 36 (25.5) | 0.263 |
| You feel that health providers do not feel comfortable offering care to transgender patients | 21 (30.9) | 39 (27.7) | 0.724 |

[1]Yes = agree or strongly agree.

Tests for significance estimated with missing data excluded using global chi-square test; Fisher's exact test for variables with cell ≤5.

childhood experiences among homeless children and adolescents, and increasing homelessness and food insecurity over the past decade [50–52]. While we found no significant association between lifetime homelessness and familial acceptance, it is notable that those who had experienced homelessness were significantly less likely to be currently living with family. Lack of family support in young adulthood, whether due to familial rejection or poverty, can have significant negative impacts for young transgender women [16,26].

There are very limited programs to support people experiencing homelessness in Peru, and fewer still that offer services for transgender people [53]. This is particularly concerning given

recent evidence of substantial discrimination against transgender people in Peru's rental market [54]. However, transgender people across Latin America are leading efforts to secure safe and adequate housing for their communities, particularly through "*casas para personas trans,*" collective dwellings shared by transgender women [55,56]. Notably, 11% of participants in this survey reported living in one of these shared homes for transgender women. These homes are especially important for young transgender women who may lack support networks in Lima, whether due to migration or rejection by their family of origin [57]. Efforts to improve housing and reduce homelessness among transgender women must acknowledge existing community social capital and social networks to build upon existing formal and informal housing solutions led by transgender women themselves.

Consistent with research in other contexts, we found that lifetime homelessness was associated with behavioural and structural vulnerability to HIV, with participants who had experienced homelessness more likely to report recent condomless anal sex, engagement in sex work (both lifetime and in the past 30 days), and experiences of violence and poor mental health [22,49,58]. For young transgender women, experiencing homelessness is both a consequence of transphobic discrimination as well as an important contributor to future poor health outcomes. Intervening to prevent homelessness early in life is critical to prevent compounding negative health and social outcomes [8]. Notably, young transgender women who had experienced homelessness were more likely to have been tested for HIV and were more likely to be willing to use PrEP, though these results were not statistically significant. As Peru works to improve access to PrEP, it is crucial to ensure transgender women, and young transgender women experiencing homelessness in particular, are not left behind in the design and implementation of these services [59].

Despite Peru's public health insurance programs for individuals living in severe poverty, 28% of participants who had experienced homelessness reported not having health insurance, compared to 18% of those who had never been homeless. Further, this study identified several other

**Table 5. Poisson regression models with robust standard errors estimating the association between homelessness and past 6-month condomless sex with a partner living with HIV or status unknown partner among young transgender women in Lima, Peru.**

|  | Model 1 PR (95% CI) | Model 2 aPR (95% CI) | Model 3 aPR (95% CI) | Model 4 aPR (95% CI) |
|---|---|---|---|---|
| **Homelessness** (Ever vs never) | 1.87 (1.37-2.56) | 1.90 (1.39-2.60) | 1.43 (1.05-1.96) | 1.19 (0.87-1.63) |
| **Age** (per 1 year increase) | – | 0.99 (0.91-1.07) | 0.97 (0.89-1.04) | 0.99 (0.92-1.07) |
| **Secondary school completion** (Ref = no secondary school) | – | 0.85 (0.60-1.21) | 1.08 (0.77-1.50) | 1.05 (0.77-1.43) |
| **Sex work** (Ref = never) | – | – | 3.76 (1.93-7.32) | 2.43 (1.25-4.73) |
| **Non-injection drug use** (Ref = no use) | – | – | 1.18 (0.85-1.65) | 0.86 (0.64-1.15) |
| **PTSD** (ref = no PTSD) | – | – | – | 1.34 (1.02-1.76) |
| **Ever any violence** (ref = never) | – | – | – | 3.78 (1.76-8.12) |

All models included N = 176 participants: N = 180 had complete data on the outcome (condomless sex in the past 6 months) and N = 4 were excluded due to missing covariate data.

Abbr: PR: Prevalence ratio; aPR: adjusted prevalence ratio; 95% CI: 95% confidence interval.

barriers to healthcare for young transgender women with experiences of homelessness, including concerns about discrimination from healthcare providers and logistical barriers such transportation, safety traveling to the health clinic, and inconvenient hours. These findings are consistent with other, commonly cited barriers to healthcare for people experiencing homelessness [60]. The significantly higher rates of anticipated discrimination by healthcare providers indicate the transgender women who have been homeless may expect intersecting discrimination based on both their transgender identity as well as their experiences of being poor and/or homeless [61]. Given the high rates of homelessness found among this population, healthcare services for all transgender women in Peru should be attentive to the needs of individuals with experiences of homelessness to improve accessibility to needed healthcare, such as HIV prevention and care. Routinely screening for homelessness in clinical settings and offering referrals to services is recommended.

These exploratory results must be understood in the context of several limitations. First, the cross-sectional design of the study limits causal inference, particularly since participants were not asked when they first experienced homelessness. We also lacked details on participants' experiences of homelessness, including what they understood as the cause, the type of homelessness (e.g., unsheltered homelessness, temporary accommodation), and the frequency and duration of homelessness. Housing instability is dynamic in nature and future research is needed to evaluate longitudinal patterns of homelessness among young transgender women, particularly given prior research demonstrating that young transgender women ages 18-24 years experience increased duration of homelessness compared to transgender women in older age groups [49,62]. Qualitative research is also warranted to explore the antecedents and consequences of homelessness for this study population. This study also had a small sample size, limiting the precision of the estimates and necessitating the use of lifetime homelessness rather than recent homelessness for modeling. The sample size was further reduced by participants opting out of HIV and STI testing, limiting the precision of the estimates for the associations between homelessness and HIV status. In addition, some scales (transgender-specific intimate partner violence, PC-PTSD) have not yet been validated in Spanish. Lastly, potential generalizability may be limited in this peer-recruited sample; findings may not generalize to young transgender women who are less connected to transgender peers.

Our study findings present some of the first estimates of experiences of homelessness and the associations with HIV vulnerabilities among transgender women in Latin America. Results underscore compounding HIV vulnerabilities faced by young transgender women who experience homelessness, including higher rates of sex work, violence, psychological distress, and barriers to healthcare access. Integrated social supports to prevent homelessness, housing support services, and tailored HIV prevention and care among young transgender women in Peru are urgently needed, in addition to ensuring services for transgender women are accessible to those who are currently homeless and those with a history of homelessness. Integrated strategies that combine housing assistance with HIV prevention and care service delivery represent an important next step forward to address structural vulnerability and HIV and STI morbidity for young transgender women in Peru.

## Supporting information

**S1 Table. HIV and STI vulnerability among young transgender women with and without experiences of homelessness (categorical homelessness).**
(DOCX)

**S2 Table. Poisson regression models with robust standard errors estimating the association between recent homelessness and past 6-month condomless sex with a partner living with HIV or status unknown partner among young transgender women in Lima, Peru.**
(DOCX)

**S1 Checklist. Inclusivity in global research questionnaire.**
(DOCX)

## Acknowledgements

We wish to thank Yahaira Chavarri, Anto Garcia, Flavia Cuenca, Mia Loarte and the young transgender women who participated in this research and shared their lived experiences with us.

## Author contributions

**Conceptualization:** Alfonso Silva-Santisteban, Sari L. Reisner, Amaya Perez-Brumer.

**Data curation:** Alfonso Silva-Santisteban.

**Formal analysis:** Dorothy Apedaile.

**Funding acquisition:** Alfonso Silva-Santisteban, Sari L. Reisner, Amaya Perez-Brumer.

**Investigation:** Alfonso Silva-Santisteban, Sari L. Reisner, Leyla Huerta, Segundo R. León, Amaya Perez-Brumer.

**Methodology:** Dorothy Apedaile, Alfonso Silva-Santisteban, Sari L. Reisner, Segundo R. León, Amaya Perez-Brumer.

**Project administration:** Alfonso Silva-Santisteban, Sari L. Reisner, Leyla Huerta, Segundo R. León.

**Supervision:** Sari L. Reisner, Amaya Perez-Brumer.

**Writing – original draft:** Dorothy Apedaile.

**Writing – review & editing:** Dorothy Apedaile, Alfonso Silva-Santisteban, Sari L. Reisner, Leyla Huerta, Segundo R. León, Amaya Perez-Brumer.

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
