## [Decision Letter · Decision Letter 0]

28 Nov 2024

PGPH-D-24-02165

Lifetime homelessness among young transgender women in Lima, Peru is associated with HIV vulnerability: Results from a cross-sectional survey

Dear Dr. Apedaile,

Thank you for submitting your manuscript to PLOS Global Public Health. After careful consideration, we feel that it has merit but does not fully meet PLOS Global Public Health’s publication criteria as it currently stands. Therefore, we invite you to submit a revised version of the manuscript that addresses the points raised during the review process.

We look forward to receiving your revised manuscript.

Kind regards,

Katia Bruxvoort, PhD

Academic Editor

Journal Requirements:

Additional Editor Comments (if provided):

Reviewers' comments:

Reviewer's Responses to Questions

**Comments to the Author**

1. Does this manuscript meet PLOS Global Public Health’s publication criteria ? Is the manuscript technically sound, and do the data support the conclusions? The manuscript must describe methodologically and ethically rigorous research with conclusions that are appropriately drawn based on the data presented.

Reviewer #1: Yes

Reviewer #2: Yes

2. Has the statistical analysis been performed appropriately and rigorously?

Reviewer #1: Yes

Reviewer #2: Yes

3. Have the authors made all data underlying the findings in their manuscript fully available (please refer to the Data Availability Statement at the start of the manuscript PDF file)?

Reviewer #1: No

Reviewer #2: Yes

4. Is the manuscript presented in an intelligible fashion and written in standard English?

Reviewer #1: Yes

Reviewer #2: Yes

5. Review Comments to the Author

Reviewer #1: The cross-sectional study investigates the relationship between homelessness and HIV vulnerability among 209 young transgender women in Lima, Peru. They found that roughly a third had experienced homelessness at some point in their lives, and 1/10 in the past 3 months. Those who had been homeless were more likely to have engaged in condomless sex, been in sex work, experienced violence, and to have suffered from PTSD. HIV prevalence was slightly higher among those with a history of homelessness than those who had never been homeless. The authors state that their findings highlight the urgent need for housing support and tailored HIV prevention and care services for young transgender women in Peru.

This study fills an important gap, and the authors highlight in their introduction that no estimates of prevalence of homelessness among transgender women were available outside North America prior to this study. This study therefore provides some of the first estimates of experiences of homelessness among transgender women in Latin America, which is hugely important.

I felt that the manuscript was well written, and the findings presented are both new and interesting. My comments focus largely on strengthening the background and justification for the analysis, further understanding the analytical approach taken to investigate the influence of lifetime homelessness, and softening the causal language used in the discussion section.

Please see my comments on each section below.

Introduction:

1. In reading the introduction, I felt that the justification for focusing on transgender women in this analysis was clear (few estimates of homelessness among transgender women, key population for HIV, often experience violence and other structural exposures that influence HIV risks). It was not immediately apparent to me why the authors chose to restrict their study to only young transgender women. Can the authors provide further information on to why this study population in particular was chosen?

2. Potential typo on page 3, line 26 with “ages 15-24”. Elsewhere in the manuscript, including in the methods, the young study population is defined using ages 16-24.

3. In the introduction, a lot of information is given about how homelessness may influence HIV vulnerabilities through restricting access to healthcare services (e.g., HIV treatment uptake and adherence, PrEP adherence, discrimination for healthcare workers, lack of health insurance etc.) and these pathways appear to be the focus, but these were not investigated in the regression analysis, which focused on links with condom use. Could the authors provide further justification/hypotheses that explain why they chose to focus on the analysis variables of interest? For example, in the introduction there is no mention of PTSD or non-injection drug use, which are important variables in the regression models.

Methods:

4. Page 5, line 77. Perhaps more information (or at least a reference, if available) on the transfeminine continuum would be useful for readers who are less familiar? I notice this information is also not provided in your previous publication on this study.

5. Page 5, lines 79-80: “Participants were recruited by peers and surveys were administered by peer survey interviewers”. From your previous paper it appears that snowball sampling was used? If so, I would specify this.

6. Page 5: Measures. In the study, questions were asked about lifetime homelessness, past 3 months homelessness, and past 30 days homelessness, as well as people’s current living situation. I would like some further information as to why the authors chose to focus on lifetime exposure as their primary exposure measure. In the discussion, it is stated that this is due to limited sample size, which presumably means too few people responded yes to questions about homelessness over shorter time periods than lifetime. Fine. My concern, however, is that the interpretation of the estimates generated with these analyses is muddled, because for some participants lifetime exposure could have been yesterday, while for others it could have been ~10 years ago or so. As such, the findings of these analyses do not seem to be particularly informative or actionable, and estimates based on more recent exposure may perhaps be more useful.

7. Page 7: Statistical analysis. Throughout, there is a lot of focus on significance testing using p<0.05 to compare differences in characteristics between the two homelessness groups. The approach of relying only on p-values is known to be misleading (See: https://pmc.ncbi.nlm.nih.gov/articles/PMC1119478/), therefore I would put let emphasis on those.

8. Complete case analysis: For the regression models, why did the authors decide the do a complete case analysis? (i.e., exclude missing data on the covariates) Did the authors attempt to account for missing data in their regression models, e.g., using the missing indicator method? (https://arxiv.org/abs/2111.00138) Additionally, does the analytical sample of 176 apply to all three regression models?

9. Regression approach: I am a bit confused about the regression approach. Can the authors explain why they chose to run three models, and sequentially incorporate different potential confounders in each? How were the variables to add in each model selected? Was selection based on statistical criteria (e.g., in stepwise selection) or based on prior causal hypotheses of these links?

10. Furthermore, I am not sure about all the variables included in the models. In particular, I am not sure about adjusting for sex work and non-injection drug use as potential confounders of the association between homelessness and condom use. Rather, I see these variables as potential mediators (and can similarly view violence and PTSD as potential mediators). Adjusting for potential mediating variables could induce collider-stratification bias and result in spurious associations if there are unmeasured confounders of the mediator and outcome (https://onlinelibrary.wiley.com/doi/10.1111/ppe.12474). Additionally, there may be the possibility that by adjusting for these variables you are blocking the pathways by which homelessness may influence the outcome, which results in the “non-significant” association in model 4. As such, I find the results of models 1 and 2 more useful than models 3 and 4. Therefore, can the authors provide further explanation for why they chose to adjust for these behavioural and psychosocial variables as confounders?

Results:

11. The authors found that 67 participants tested positive for HIV. Is there any indication of how many of these participants knew they were living with HIV before the study started?

12. I find that the descriptive results in this manuscript are particularly interesting, and I like the authors focus on these findings in the results section. It is an awful lot of information however, so it would be nice, if possible, to see some of it in a figure or two rather than only in tables.

13. As stated in a previous comment, I would focus less on p-values for comparing differences, and just report those that seem most interesting. Even if the p-value is <0.0001 this does not necessarily suggest a difference between the groups and could just be due to chance, particularly as you are comparing lots of groups.

14. Page 14, lines 241-243. I think the word “without” is missing from this sentence.

Discussion:

15. My main concern in the discussion is the use of strong causal language to describe what were only observational cross-sectional associations in your study. On page 18, the paragraph beginning with “Consistent with research in other contexts…” the authors state that “we found that lifetime homelessness increased behavioural and structural vulnerability to HIV through increased likelihood of recent condomless anal sex, increased engagement in sex work, and increased likelihood of experiencing violence and poor mental health”. However, the analysis did not show this, it demonstrated cross-sectional links between homelessness and condom use, when adjusting for different variables, and the associations were not consistent in the 4 models run. As this was a cross-sectional analysis, we are not able to infer causality or exclude the possibility of reverse causality in relationships. I think the authors need to be more attentive to this in their discussion and in places soften the language to highlight the observed links without implying a causal or temporal direction e.g., “those who ever experienced homelessness were more likely to have engaged in sex work” rather than “homelessness increased engagement in sex work” etc. The authors acknowledge in the limitations that the design of the study limits causal inference, but the language used through the discussion implies they think the findings of the analyses represent causal relationships.

Reviewer #2: Introduction:

It would help to better anchor your intro in terms of the extent to which you are starting with global data then zooming in on peru.

Methods:

Employment status: to what extent does that consider sex work as work? Please clarity

To what extent have these scales been validated in Spanish?

Please provide v brief context on Peruvian health system to better locate the extent to which health insurance is universal, employment specific etc - in general read through with this lens that readers may not have in depth knowledge of the country specific health system so consider spelling out / providing definintions and context even if only v briefly.

In general it was exceptionally written!

6. PLOS authors have the option to publish the peer review history of their article (what does this mean? ). If published, this will include your full peer review and any attached files.

**Do you want your identity to be public for this peer review?** For information about this choice, including consent withdrawal, please see our Privacy Policy .

Reviewer #1: **Yes: ** James Stannah

Reviewer #2: No

---

## [Editor Report · Decision Letter 1]

11 Feb 2025

Lifetime homelessness among young transgender women in Lima, Peru is associated with HIV vulnerability: Results from a cross-sectional survey

PGPH-D-24-02165R1

Dear Ms Apedaile,

We are pleased to inform you that your manuscript 'Lifetime homelessness among young transgender women in Lima, Peru is associated with HIV vulnerability: Results from a cross-sectional survey' has been provisionally accepted for publication in PLOS Global Public Health. I have assessed the revised manuscript and made the decision to accept without inviting re-review. 

Best regards,

Katia Bruxvoort, PhD

Academic Editor
